# Joint Power Control and Phase Shift Design for Future PD-NOMA IRS-Assisted Drone Communications under Imperfect SIC Decoding

**DOI:** 10.3390/s22228603

**Published:** 2022-11-08

**Authors:** Saddam Aziz, Muhammad Irshad, Kallekh Afef, Heba G. Mohamed, Najm Alotaibi, Khaled Tarmissi, Mrim M. Alnfiai, Manar Ahmed Hamza

**Affiliations:** 1Centre for Advances in Reliability and Safety, New Territories, Hong Kong; 2Department of Electronics and Information Engineering, The Hongkong Polytechnic University, Hung Hom, Hong Kong; 3Department of Mathematics, College of Science & Arts at Mahayil, King Khalid University, Mohail Asser, Abha 62529, Saudi Arabia; 4Department of Electrical Engineering, College of Engineering, Princess Nourah bint Abdulrahman University, P.O. Box 84428, Riyadh 11671, Saudi Arabia; 5Prince Saud AlFaisal Institute for Diplomatic Studies, Riyadh 11553, Saudi Arabia; 6Department of Computer Sciences, College of Computing and Information System, Umm Al-Qura University, Mecca 24382, Saudi Arabia; 7Department of Information Technology, College of Computers and Information Technology, Taif University, P.O. Box 11099, Taif 21944, Saudi Arabia; 8Department of Computer and Self Development, Preparatory Year Deanship, Prince Sattam Bin Abdulaziz University, Al-Kharj 11671, Saudi Arabia

**Keywords:** intelligent reflecting surfaces, drones, non-orthogonal multiple access, imperfect successive interference cancellation decoding, spectral efficiency optimization

## Abstract

Intelligent reflecting surfaces (IRS) and power-domain non-orthogonal multiple access (PD-NOMA) have recently gained significant attention for enhancing the performance of next-generation wireless communications networks. More specifically, IRS can smartly reconfigure the incident signal of the source towards the destination node, extending the wireless coverage and improving the channel capacity without consuming additional energy. On the other side, PD-NOMA can enhance the number of devices in the network without using extra spectrum resources. This paper proposes a new optimization framework for IRS-enhanced NOMA communications where multiple drones transmit data to the ground Internet of Things (IoT) devices under successive interference cancellation errors. In particular, the power budget of each drone, PD-NOMA power allocation of IoT devices, and the phase shift matrix of IRS are simultaneously optimized to enhance the total spectral efficiency of the system. Given the system model and optimization setup, the formulated problem is coupled with three variables, making it very complex and non-convex. Thus, this work first transforms and decouples the problem into subproblems and then obtains the efficient solution in two steps. In the first step, the closed-form solutions for the power budget and PD-NOMA power allocation subproblem at each drone are obtained through Karush–Kuhn–Tucker (KKT) conditions. In the second step, the subproblem of efficient phase shift design for each IRS is solved using successive convex approximation and DC programming. Numerical results demonstrate the performance of the proposed optimization scheme in comparison to the benchmark schemes.

## 1. Introduction

Drone communications are expected to play a significant role in future mobile networks, providing high-speed and on-demand wireless connectivity [1]. Because of its distinct channel characteristics, accurate channel modeling of drone-aided air–ground communication is critical for network performance analysis and system design [2]. Drones can be used to assist the ground base station (BS) in serving users with high data traffic and overloaded cases due to their flexibility in deployment and cost-effectiveness [3,4]. Furthermore, drones can be combined with key physical layer technologies to achieve higher capacity, such as power-domain non-orthogonal multiple access (PD-NOMA) [5], millimeter wave (mmWave) communications [6], backscatter communications [7], intelligent reflecting surfaces (IRS) [8], mobile edge computing [9,10,11], cognitive radio [12], terahertz communication [13] and so on.

Traditional orthogonal multiple access (OMA) methods, such as time division multiple access (TDMA) and orthogonal frequency division multiple access (OFDMA), only support a single user per orthogonal resource block [14]. The following straightforward example demonstrates the OMA’s spectral inefficiency. For the sake of fairness, suppose that one user who is experiencing very poor channel conditions nevertheless needs to be serviced because they have high priority data or have not been served for a long time [15]. As a result of this user’s poor channel conditions, OMA guarantees that they will consume all of one of the few available bandwidth resources. This obviously reduces the system’s overall spectrum efficiency and throughput [16]; whereas PD-NOMA guarantees that the user with poor channel circumstances is served, it also allows users with better channel conditions to share the same bandwidth without negatively impacting the weak user’s experience [17]. Therefore, if user fairness is to be ensured, PD-NOMA’s system throughput can be much higher than that of OMA. Academic and corporate research has shown that NOMA can efficiently handle vast connectivity, which is critical for assuring that future wireless networks can support Internet of Things (IoT) features [18].

In recent years, intelligent reflecting surface (IRS) has emerged as a promising technique to reconfigure the propagation environments and enhance network performance, thanks to the availability of massive low-cost passive reflecting elements [19]. IRS requires much less energy consumption and can be easily deployed on building facades, ceilings, and walls in comparison to existing techniques like active relay and backscattering communication. In order to improve capacity, energy efficiency, and physical layer security, IRS has recently been considered for use in terrestrial networks [20]. To achieve various communication goals, various network designs can jointly optimize the phase shifts of reflecting elements and the transmission control of transceivers [21].

In spite of drone communication’s many benefits, air-to-ground routes may be obstructed by the surrounding terrain and environment [22]. In addition, when eavesdropping is present, it is possible that genuine users’ data security will be compromised. IRS can be used in drone-aided air–ground networks to combat these challenges by creating a more favorable propagation environment and enhancing the quality of communication for target users [23]. By carefully crafting the passive beamforming, IRS can simultaneously cancel out the unwanted signals to decrease interference and prevent adversary eavesdropping. Recently, research has emerged that combines drone communications with IRS to boost the efficiency of air–ground networks [24]. In particular, IRS allows for increased drone coverage, allowing for the support of a wide variety of user QoS needs. IRS deployed on a mobile drone allows for greater deployment flexibility and a larger signal reflection range than when put on a stationary structure. Accordingly, IRSs can also be placed onto building walls to assist the signal delivery from drone to ground IoT devices.

As is well known, the channel power gains of various users in PD-NOMA transmission are crucial for successful SIC decoding at receiver side. However, PD-NOMA faces new challenges when implemented in IRS-assisted drone communication systems, where the channel response can be artificially altered by adjusting the phase shifts. Although there exist numerous research works on IRS and PD-NOMA, to the best of authors knowledge there is no work that exists on the sum capacity maximization of PD-NOMA IRS-assisted multiple drone communications under imperfect SIC decoding.

Following the characteristics of drones, IRS, and NOMA mentioned above, it is critical to integrate these technologies to improve the performance of existing wireless communications networks. Therefore, this work proposes a new optimization scheme for NOMA IRS-assisted drone communications networks, where multiple drones share the same frequency and communicate with ground IoT devices using the NOMA protocol. Due to large objects in urban areas, some IoT devices face large-scale fading. Hence, we consider multiple IRS systems mounted strategically on walls to assist signal delivery from drones to IoT devices. The objective of this is to maximize the total achievable capacity of the system under imperfect SIC decoding. In particular, the proposed framework simultaneously optimizes the total power of each drone, transmits the power of IoT devices according to NOMA, and optimizes the phase shift design of IRS subject to the minimum capacity requirements of IoT devices. The problem of total achievable capacity maximization is formulated as non-convex; thus, it is very complex to get a joint optimal solution. To make the optimization problem tractable and reduce the overall complexity, we first decouple the optimization problem into subproblems. Then, we exploit KKT conditions to calculate the power allocation of drones while adopting successive convex approximation and DC programming to obtain an efficient phase shift design at IRS. To validate the proposed framework, we also provide simulation results which show the benefits of the proposed scheme. The main contributions of this work can be summarized as follows.

A downlink PD-NOMA system is considered that consists of multiple drones, multiple IRSs, and IoT devices, where each drone in its coverage area communicates with IoT devices through direct and IRS-assisted links. Due to large objects in urban areas, IRSs are mounted on strategic positions to assist the signal delivery from drones to IoT devices. To maximize the spectral efficiency of the system, each drone shares the same spectrum resources. Thus, IoT devices in the coverage area of one drone receive interference from neighboring drones. Besides that, IoT devices in the same coverage area also cause PD-NOMA interference. Moreover, interference due to imperfect SIC also exists in the system. Therefore, the objective of this framework is to enhance the sum capacity of the system through efficient resource allocation.The problem is formulated to enhance the sum capacity maximization of the system subject to quality of services and other practical constraints. In particular, the proposed approach simultaneously optimizes the transmit power budget of drones, PD-NOMA power allocation for IoT devices, and phase shift design of IRSs. Due to the non-convex nature of the formulated problem, computing optimal solution directly is very challenging. To make it tractable and reduce the complexity, we first divide the joint problem into subproblems and then obtain an efficient solution. For power allocation subproblem, we adopt a Lagrangian method based on KKT conditions where dual variables are updated iteratively. Next, for efficient phase shift design, we employ successive convex approximation and the DC programming method.To validate the proposed solution, numerical results are provided to check the system’s performance with respect to different optimization variables. For better analysis, we compare the proposed solution with benchmark solutions such as a solution with perfect SIC decoding, a solution without IRS, and a solution where only long-distance IoT device signals can be assisted by IRS. The results demonstrate that the proposed approach outperforms the benchmark solutions in the sum capacity maximization of the system. Moreover, our approach contain very low complexity and converges within a few iterations.

The remaining paper can be structured as follows: Section 2 will provide a system model and formulate a total achievable maximization problem. Section 3 studies the process of the proposed solution, including the decoupling of the problem, transformations, and closed-form expressions. Section 4 provides and discusses numerical results obtained through Monte Carlo simulations. Finally, we conclude this work in Section 5. Different notations used in our work can be found in Table 1.

### Recent Literature

Due to their potential usefulness in a wide variety of civilian contexts, drones have recently been projected to grow into a USD 55 billion industry worldwide by 2027 [25]. Drones can be used in a wide variety of situations due to their rapid deployment on demand, low cost of operation and maintenance, and flexible and controllable maneuverability [26]. Some examples include monitoring traffic in real time, precision agriculture, remote sensing, communication relaying, and wireless coverage. Due to their mobility, drones can establish reliable line-of-sight (LoS) air–ground links to IoT devices on the ground, which is especially useful in the drone wireless communications paradigm. This solves a problem with conventional fixed-infrastructure wireless networks by providing improved wireless coverage while consuming less power. As a result, there has been extensive study devoted to the development of systems for efficient path planning/deployment and resource management in drone IoT networks [27,28,29]. However, the vast majority of previous studies have focused on the sub-6 GHz bands of the microwave spectrum for drone communications. Because these frequencies are already being used extensively by older wireless technologies, we are facing a “spectrum crunch” [30]. This is why future drone IoT networks are searching for designs of drone communications systems on other frequency bands, such as the promising terahertz (THz) bands.

Recently, IRS has been integrated to drone communications to further enhance the air–ground communications. IRS-assisted drone communications [31,32,33,34,35,36] and the references therein have seen a rise in popularity as of late. For instance, in [31], a drone was considered as an aerial user equipment, and an IRS installed on a nearby building was used to enhance the drone’s communication channel capacity. Received power was shown to increase with increasing the number of IRS elements, as shown by the authors of the article. They also showed that IRSs are more useful at higher altitudes for the drone. However, the maximum gain was reached once the drone followed the main lobe of the BS antenna. A conclusion was reached: the BS’s down-tilted antenna pattern determines where the drone and IRS should be placed for maximum efficiency. In [32], a drone’s IRS was powered via energy harvesting by the non-reflected portion of an impinging wave. Multiple antennas at the BS with beamforming towards the IRS were considered, and the propagation environment was modeled with reinforcement learning to maintain a LoS connection between the drone and the IRS while the ground user equipment was in motion. This research demonstrates that, even at low drone altitudes, IRS can significantly increase spectral efficiency. The authors in [33] made the same assumption, in which numerous IRSs were installed on the outsides of nearby building walls. Because the drone has multiple antennas, the ground user can pick up signals from the drone directly as well as from the reflection of the IRSs. In order to achieve maximum received power at the ground user, it was necessary to optimize both the passive beamforming at the IRSs and the trajectory of the drone. As shown on this premise that the amount of power harvested grows exponentially with the number of reflecting elements of installed IRS. In addition, comparable to the system model considered in [31], but with a single IRS, was examined by the authors of [34]. Note that the drone’s active beamforming, the IRS’s passive beamforming, and the drone’s trajectory were all collaboratively adjusted to maximize the average throughput at the ground user. It was demonstrated that when compared to a scheme in which decoupled optimization is carried out, the average rate was greater when joint optimization was implemented.

Similarly, to further improve the signal quality, a user equipment combined transmissions from a drone equipped with an IRS [35]. There were three approaches taken to accomplish this goal: using only the drone, using only the IRS, and having the IRS aid the drone. Both the ergodic capacity and the outage probability could be written in closed form for this configuration. It was demonstrated that the IRS-only mode is more power efficient for LoS communication and when the drone is located closer to the user [37]. To improve communication between a ground station and mobile devices, researchers in [36] deployed an IRS on a drone. To maximize the signal-to-noise ratio over a rectangular coverage area, aerial IRS involves optimizing the drone deployment, BS beamforming vector, and IRS passive beamforming [38], whereas the LoS communication channels are used for all communications links [39]. The optimal drone placement was carried out at the user equipment site based on the measured height difference between the user equipment and the drone. Assuming a rectangular area small enough to be covered by the IRS array response, the authors found that the array gain scales quadratically with the number of reflection elements. In all other aspects, the gain of an IRS array increases proportionally with the number of elements, as the area does. Other than that study, all of the above research considered either a distance-based path loss model with Rician fading or a dual-slope height path loss model with spatial channel models for the drone to link with user equipment [40]. In [35], a probability of LoS, path loss component, and a path loss exponent that depends on the elevation angle was used to optimize the BS-to-drone and drone-to-user equipment links. It is possible that the LoS probability used in this analysis is not applicable to the BS to drone link because it is based on the drone-to-user equipment link [41]. Specifically, they employed the LoS probability model for the BS-to-drone link that was created and presented in [42]. The authors in [43] proposed a cooperative non-terrestrial network to investigate the outage probability and bit error rate.

## 2. System Model and Problem Formulation

A downlink PD-NOMA IRS-assisted drone communication scenario is considered where multiple drones communicate with IoT devices in an urban area using multiple IRSs, as shown in Figure 1. Let us denote the number of drones as *U* such that {u=1,2,3,⋯,u,⋯,U}. In the considered model, we assume that each drone covers a geographical area and transmits data to two IoT devices. For simplicity, this work considers that each drone communicates with two IoT devices simultaneously. However, this model can be easily extended to large IoT devices by considering multiple resource blocks at each drone. In such a case, drones can efficiently share all resource blocks so that each resource block accommodates multiple PD-NOMA IoT devices. To enhance the spectral efficiency, all the drones share the same spectrum, hence cause co-channel interference with each other [44]. Let us assume that *l* and *m* are the two IoT devices associated with drone *u*, where u∈U. Without loss of generality, IoT device *l* has strong channel conditions due to the line-of-site (LoS) connection with drone *u*. However, the IoT device *m*’s connection with drone *u* is blocked by large objects, hence there is no direct connection from the drone *u* to IoT device *m*. To address this issue and enhance the overall system performance, we consider an IRS in the coverage area of each drone that is mounted on a strategic position for the delivery of the signal of a drone *u* to IoT device *l* and IoT device *m*. The IRS consists of *V* passive elements such that its diagonal matrix for reflecting the signal of drone *u* can be expressed as:(1)$$\begin{array}{c} \Theta_{u}=\mathrm{diag}\left\{\psi_{1,u}e^{j\theta_{1,u}},\psi_{2,u}e^{j\theta_{2,u}},\cdots ,\psi_{V,u}e^{j\theta_{V,u}}\right\} \end{array}$$
where $\psi_{v,u}\in \left[0,1\right]$ denotes the amplitude of passive reflection and $\theta_{v,u}\in \left[0,2\pi \right]$ is the phase shift of passive element *v* at IRS. Next, we assume that $x_{u}$ is the transmitted signal of drone *u* to IoT device *l* and IoT device *m* which can be written as:(2)$$\begin{array}{c} x_{u}=\sqrt{Q_{u}\varrho_{l,u}}x_{l,u}+\sqrt{Q_{u}\varrho_{m,u}}x_{m,u},\;\forall u\in U, \end{array}$$
where $Q_{u}$ is the transmit power of drone *u*, $\varrho_{l,u}$ denotes the power allocation coefficient of IoT device *l*, $\varrho_{m,u}$ shows the power allocation coefficient of IoT device *m*, $x_{l,m}$ denotes the unit power signal of IoT device *l*, and $x_{m,u}$ shows the unite power signal of IoT device *m*. In the proposed model, the channel between the drone and IoT device *l* and between drone and the IRS is LoS based and can be modeled as:(3)$$\begin{array}{c} h_{\kappa ,u}=\frac{\vartheta_{0}}{∥\zeta_{u}-\upsilon_{\kappa }{∥}^{2}+H^{2}}, \end{array}$$
where $\vartheta_{0}$ shows the reference channel gain over 1 meter of distance, ζ is the 2D coordinates of each drone *u* such that $\zeta_{u}\in \left\{X_{u},Y_{u}\right\}$, and $\upsilon_{\kappa }$ denotes the location of the IRS and IoT device at horizontal plane, where κ∈{l,v}, and *H* represents the fixed altitude of drone *u*. In this work, we consider that the trajectory of drones has already been calculated before the proposed optimization framework. The optimal trajectory of drones can further enhance the system performance; however, it is beyond the scope of this work. Next, the channel between an IRS and IoT devices can be modeled as:(4)$$\begin{array}{c} g_{\iota ,u}=G_{\iota ,u}\times D_{\iota ,u}^{\frac{\phi }{2}}, \end{array}$$
where $G_{\iota ,u}$ denotes the Rayleigh fading coefficient and $D_{\iota ,u}$ is the distance between IRS and IoT device ι, where ι∈{l,m} and ϕ shows the path loss exponent. The signal that IoT device *l* receives from drone *u* through direct and IRS links can be written as:(5)$$\begin{array}{c} y_{l,u}=\left(h_{l,u}+h_{v,u}\Theta_{u}g_{l,v,u}^{H}\right)x_{u}+\sum_{u^{\prime }=1,u^{\prime }\neq u}^{U}\left(h_{l,u^{\prime }}+h_{v,u^{\prime }}\Theta_{u}g_{l,v,u^{\prime }}^{H}\right)\sqrt{Q_{u^{\prime }}\varrho_{l,u^{\prime }}}x_{l,u^{\prime }}+\omega_{l,u}, \end{array}$$
where the first segment in (Equation 5) is the desired superimposed signal at IoT device *l* through both direct and indirect links, the second segment shows the co-channel interference from other drones, and the third segment is the additive white Gaussian noise with zero mean and $\sigma^{2}$ variance. Similarly, the signal that IoT device *m* receives through IRS can be stated as:(6)$$\begin{array}{c} y_{m,u}=h_{v,u}\Theta_{u}g_{m,v,u}^{H}+\sum_{u^{\prime }=1,u^{\prime }\neq u}^{U}\left(h_{m,u^{\prime }}+h_{v,u^{\prime }}\Theta_{u}g_{m,v,u^{\prime }}^{H}\right)\sqrt{Q_{u^{\prime }}\varrho_{m,u^{\prime }}}x_{m,u^{\prime }}+\omega_{m,u}, \end{array}$$
where the first segment is the desired signal at IoT device *m* through IRS only, the second segment depicts the co-channel interference from other drones, and the last segment denotes the additive white Gaussian noise with zero mean and $\sigma^{2}$, respectively.

Following the PD-NOMA principle, the IoT device *l* associated with drone *u* can apply SIC to decode the signal of IoT device *m* before decoding its own signal. However, we assume that IoT device *l* cannot always decode it successfully. Thus, we consider errors during the decoding process of the superimposed signal. Given the above observation, we express the achievable data rate of IoT device *l* and IoT device *m* as:(7)$$\begin{array}{c} C_{l,u}=\log_{2}\left(1+\frac{Q_{u}\varrho_{l,u}\left(\right|h_{l,u}{|}^{2}+|h_{v,u}\Theta_{u}g_{l,v,u}^{H}{|}^{2})}{Q_{u}\left(1-\varrho_{l,u}\right)\left(\right|h_{l,u}{|}^{2}+|h_{v,u}\Theta_{u}g_{l,v,u}^{H}{|}^{2})\delta +\Gamma_{l,u^{\prime }}+\sigma^{2}}\right), \end{array}$$
(8)$$\begin{array}{c} C_{m,u}=\log_{2}\left(1+\frac{Q_{u}\varrho_{l,u}\left(1-\varrho_{l,u}\right){\left|h_{v,u}\Theta_{u}g_{m,v,u}^{H}\right|}^{2}}{Q_{u}\varrho_{l,u}{\left|h_{v,u}\Theta_{u}g_{m,v,u}^{H}\right|}^{2}+\Gamma_{m,u^{\prime }}+\sigma^{2}}\right), \end{array}$$
where $1-\varrho_{l,u}=\varrho_{m,u}$. The nominator in (7) is the signal gain received from drone through direct and IRS-assisted link. The denominator in (7) denotes the PD-NOMA interference, imperfect SIC interference, and $\Gamma_{l,u^{\prime }}=\sum_{u^{\prime }=1,u^{\prime }\neq u}(|h_{l,u^{\prime }}{|}^{2}+|h_{v,u^{\prime }}\Theta_{u^{\prime }}g_{l,v,u^{\prime }}^{H}{|}^{2})Q_{u^{\prime }}\varrho_{l,u^{\prime }}$ is the co-channel interference from other drones and $\sigma^{2}$ shows the variance. Accordingly, the nominator in (8) denotes the signal gain received from drone through direct and IRS-assisted links and the denominator shows the PD-NOMA interference and co-channel interference.

This work seeks to maximize the spectral efficiency of the proposed PD-NOMA IRS-assisted drone communications with imperfect SIC decoding. This can be achieved by optimizing multiple variables, i.e., the power budget of each drone, the PD-NOMA power allocation of IoT devices, and the phase shift design of every IRS in the system. Mathematically, this framework can be formulated in the following problem as:(9)$$\begin{array}{cc} (P):\; & \underset{\left(\varrho_{u},Q_{u},\Theta_{u}\right)}{\mathrm{maximize}}\;\;\sum_{u=1}^{U}\left(C_{l,u}+C_{m,u}\right) \end{array}$$$$\begin{array}{cc} s.t. & \left\{\begin{array}{c} W1:C_{l,u}\geq C_{min},\;\forall u,\\ W2:C_{m,u}\geq C_{min},\;\forall u,\\ W3:0\leq Q_{u}\leq P_{max},\;\forall u,\;\\ W4:\theta_{v,u}\in \left\{0,2\pi \right\},\;\forall k,u,\\ W5:|\psi_{v,u}|=1,\;\forall k,u,\\ W6:0\leq \varrho_{l,u}\leq \frac{1}{2},\;\forall n, \end{array}\right. \end{array}$$
where (9) is the objective function for sum capacity maximization. Constraints W1 and W2 ensure the minimum spectral efficiency of IoT device *l* and IoT device *m*, where $C_{min}$ is the threshold. Constraint W3 controls the transmission power of each drone, where $P_{max}$ shows the maximum power budget. Constraints W4 and W5 design an efficient phase shift for IRS while constraint W1 distributes the power among PD-NOMA IoT devices.

## 3. Proposed Optimization Solution

In this section, we provide an efficient solution for the formulated problem in (Equation 18). In particular, we first calculate the total power budget of each drone and PD-NOMA power allocation of IoT devices. Then, we design the efficient phase shift of each IRS.

### 3.1. Efficient Power Allocation

For any given phase shift design at IRS, the problem in (Equation 18) can be effectively updated as:(10)$$\begin{array}{cc} (P1):\; & \underset{\left(\varrho_{u},Q_{u}\right)}{\mathrm{maximize}}\;\;\sum_{u=1}^{U}\left(C_{l,u}+C_{m,u}\right) \end{array}$$$$\begin{array}{cc} s.t. & \left\{\begin{array}{c} W1,W2,W3,W6, \end{array}\right. \end{array}$$
where (P1) is the problem of drone power allocation. Next, we define the Lagrangian function as:(11)$$\begin{array}{cc} L & =-\sum_{u=1}^{U}\left(\left(C_{l,u}+C_{m,u}\right)\right)+\sum_{u=1}^{U}\lambda_{l,u}\left(C_{min}-C_{l,u}\right)+\sum_{u=1}^{U}\lambda_{m,u}\left(C_{min}-C_{m,u}\right)\\ & +\sum_{u=1}^{U}\lambda_{u}\left(Q_{u}-P_{max}\right)+\sum_{u=1}^{U}\mu_{l,u}\left(\varrho_{l,u}-0.5\right), \end{array}$$
where $\lambda \in \{\lambda_{l,u},\lambda_{m,u},\lambda_{u},\mu_{l,u}\}$ denotes the Lagrangian multipliers. Now, we employ KKT conditions by calculating the derivations of L with respect to $\varrho_{l,m}$ and $Q_{u}$. For $\varrho_{l,m}$, we obtain the following solution:$$\begin{array}{cc} \; & \varrho_{l,m}^{6}{(\left(\delta -1\right)}^{2}\delta^{2}\mu_{l,u}(|h_{l,u}{|}^{2}+|h_{v,u}{|}^{2}\Theta_{u}|g_{l,v,u}{|}^{2}{)}^{4}(|h_{m,u}{|}^{2}+|h_{v,u}{|}^{2}\Theta_{u}|g_{m,v,u}{|}^{2}{)}^{2}Q_{u}^{6}) \end{array}$$
(12)$$\begin{array}{c} +\varrho_{l,u}^{5}\gamma_{5}+\varrho_{l,u}^{4}\gamma_{4}+\varrho_{l,u}\gamma_{3}+\varrho_{l,u}^{2}\gamma_{2}+\varrho_{l,u}\gamma_{1}+\gamma_{0}=0, \end{array}$$
where the values of $\gamma_{0}$, $\gamma_{1}$, $\gamma_{2}$, $\gamma_{3}$, $\gamma_{4}$, and $\gamma_{5}$ can be found in Appendix A. It can be observed that (Equation 12) is a polynomial of order six that can be efficiently solved using a mathematical/polynomial solver. For $Q_{u}$, we can calculate its partial derivation as:(13)$$\begin{array}{cc} \; & Q_{u}^{4}(\left(\varrho_{l,u}-1\right)\varrho_{l,u}\left(\left(\delta -1\right)-\delta \right)\delta (|h_{l,u}{|}^{2}+|h_{v,u}{|}^{2}\Theta_{u}|g_{l,v,u}{|}^{2}{)}^{2}(|h_{m,u}{|}^{2}+|h_{v,u}{|}^{2}\Theta_{u}|g_{m,v,u}{|}^{2}{)}^{2}\lambda_{u})+\\ & +Q_{u}^{3}\varphi_{3}+Q_{u}^{2}\varphi_{2}+Q_{u}^{1}\varphi_{1}+\varphi_{0}, \end{array}$$
where the values of $\varphi_{0}$, $\varphi_{1}$, $\varphi_{2}$, and $\varphi_{3}$ are shown in Appendix B. During computing $\varrho_{l,u}$ and $Q_{u}$, the Lagrangian multipliers are updated as:(14)$$\begin{array}{c} \lambda_{l,u}^{\left(t+1\right)}={\left[\lambda_{l,u}\left(t\right)+\chi \left(C_{min}-C_{l,u}\right)\right]}^{+} \end{array}$$
(15)$$\begin{array}{c} \lambda_{m,u}^{\left(t+1\right)}={\left[\lambda_{m,u}\left(t\right)+\chi \left(C_{min}-C_{m,u}\right)\right]}^{+} \end{array}$$
(16)$$\begin{array}{c} \lambda_{u}^{\left(t+1\right)}={\left[\lambda_{u}\left(t\right)+\chi \left(Q_{u}-P_{max}\right)\right]}^{+} \end{array}$$
(17)$$\begin{array}{c} \mu_{l,u}^{\left(t+1\right)}={\left[\mu_{l,u}\left(t\right)+\chi \left(\varrho_{l,u}-0.5\right)\right]}^{+} \end{array}$$
where *t* is the iteration index, χ denotes the step size, and ${\left[\Psi \right]}^{+}=\max\Psi ,0$. In each *t*, the proposed scheme iteratively updated until the values of the Lagrangian and optimization variables converge.

### 3.2. Efficient Phase Shift Design

In this subsection, we investigate efficient phase shift design at each IRS. For a given power allocation scheme, the problem of phase shift design can be expressed as:(18)$$\begin{array}{cc} (P2):\; & \underset{\left(\Theta_{u}\right)}{\mathrm{maximize}}\;\;\sum_{u=1}^{U}\left(C_{l,u}+C_{m,u}\right) \end{array}$$$$\begin{array}{cc} s.t. & \left\{\begin{array}{c} W1,W2,W4,W5, \end{array}\right. \end{array}$$
where the above problem is the efficient phase shift design at IRS. To reduce the complexity of the above problem and obtain an efficient solution, we exploit the successive convex approximation method. Based on this method, the rate of IoT device *l* and *m* associated with drone *u* can be effectively written as:(19)$$\begin{array}{c} C_{l,u}=\Xi_{l,u}\log_{2}\left(SINR_{l,u}\right)+\Pi_{l,u}, \end{array}$$
(20)$$\begin{array}{c} C_{m,u}=\Xi_{m,u}\log_{2}\left(SINR_{m,u}\right)+\Pi_{m,u}, \end{array}$$
where $\Xi_{l,u}$, $\Xi_{m,u}$, $\Pi_{l,u}$, and $\Pi_{m,u}$ are, respectively, give as:(21)$$\begin{array}{c} \Xi_{l,u}=\frac{SINR_{l,u}}{1+SINR_{l,u}}, \end{array}$$
(22)$$\begin{array}{c} \Xi_{m,u}=\frac{SINR_{m,u}}{1+SINR_{m,u}}, \end{array}$$
(23)$$\begin{array}{c} \Pi_{l,u}=\log_{2}\left(1+SINR_{l,u}\right)-\frac{SINR_{l,u}}{1+SINR_{l,u}}, \end{array}$$
(24)$$\begin{array}{c} \Pi_{m,u}=\log_{2}\left(1+SINR_{m,u}\right)-\frac{SINR_{m,u}}{1+SINR_{m,u}}, \end{array}$$

Note that $SINR_{l,u}$ and $SINR_{m,u}$ are the signal-to-interference-plus-noise ratios of IoT device *l* and *m* and can be expressed as:(25)$$\begin{array}{c} SINR_{l,u}=\frac{Q_{u}\varrho_{l,u}(|h_{l,u}{|}^{2}+|h_{v,u}\Theta_{u}g_{l,v,u}^{H}{|}^{2})}{Q_{u}\left(1-\varrho_{l,u}\right)(|h_{l,u}{|}^{2}+|h_{v,u}\Theta_{u}g_{ł,v,u}^{H}{|}^{2})\delta +\Gamma_{l,u^{\prime }}+\sigma^{2}}, \end{array}$$
(26)$$\begin{array}{c} SINR_{m,u}=\frac{Q_{u}\varrho_{l,u}\left(1-\varrho_{l,u}\right){\left|h_{v,u}\Theta_{u}g_{m,v,u}^{H}\right|}^{2}}{Q_{u}\varrho_{l,u}{\left|h_{v,u}\Theta_{u}g_{m,v,u}^{H}\right|}^{2}+\Gamma_{m,u^{\prime }}+\sigma^{2}}, \end{array}$$

Next, we can efficiently transform the rate of IoT device *l* and IoT device *m* as:(27)$$\begin{array}{cc} C_{l,u} & =\Xi_{l,u}(\log_{2}(Q_{u}\varrho_{l,u}(|h_{l,u}{|}^{2}+|h_{v,u}\Theta_{u}g_{l,v,u}^{H}{|}^{2}))\\ & -\log_{2}(Q_{u}\left(1-\varrho_{l,u}\right)(|h_{l,u}{|}^{2}+|h_{v,u}\Theta_{u}g_{l,v,u}^{H}{|}^{2})\delta +\Gamma_{l,u^{\prime }}+\sigma^{2}))+\Pi_{l,u}, \end{array}$$
(28)$$\begin{array}{cc} C_{m,u} & =\Xi_{m,u}(\log_{2}(Q_{u}\varrho_{l,u}\left(1-\varrho_{l,u}\right)|h_{v,u}\Theta_{u}g_{m,v,u}^{H}{|}^{2}))\\ & -\log_{2}(Q_{u}\varrho_{l,u}|h_{v,u}\Theta_{u}g_{m,v,u}^{H}{|}^{2}+\Gamma_{m,u^{\prime }}+\sigma^{2}))+\Pi_{m,u}, \end{array}$$

Now, let us assume that $\Omega_{u}=\Theta_{u}\Theta_{u}^{H}$ and $\Psi_{l,v,u}={\left|h_{v,u}g_{m,v,u}^{H}\right|}^{2}$, where $\Omega_{u}\geq 1$, $\Psi_{l,v,u}\geq 1$ and $\mathrm{rank}(\Omega_{u})=1$, $\mathrm{rank}(\Psi_{l,v,u})=1$, respectively. Then, $\xi_{l,1}\left(\Omega_{u}\right)$ and $\xi_{l,2}\left(\Omega_{u}\right)$ can be written as:(29)$$\begin{array}{c} \xi_{l,1}\left(\Omega_{u}\right)=\log_{2}(Q_{u}\varrho_{l,u}\left(\right|h_{l,u}{|}^{2}+\Omega_{u}\Psi_{l,v,u})), \end{array}$$
(30)$$\begin{array}{c} \xi_{l,2}\left(\Omega_{u}\right)=\log_{2}(Q_{u}\left(1-\varrho_{l,u}\right)\left(\right|h_{l,u}{|}^{2}+\Omega_{u}\Psi_{l,v,u})\delta +\Gamma_{l,u^{\prime }}+\sigma^{2}), \end{array}$$

Accordingly, $\xi_{m,1}\left(\Omega_{u}\right)$ and $\xi_{m,2}\left(\Omega_{u}\right)$ are given as:(31)$$\begin{array}{c} \xi_{m,1}\left(\Omega_{u}\right)=\log_{2}\left(Q_{u}\varrho_{l,u}\left(1-\varrho_{l,u}\right)\Omega_{u}\Psi_{m,v,u}\right)), \end{array}$$
(32)$$\begin{array}{c} \xi_{m,2}\left(\Omega_{u}\right)=\log_{2}\left(Q_{u}\varrho_{l,u}\Omega_{u}\Psi_{m,v,u}+\Gamma_{m,u^{\prime }}+\sigma^{2}\right), \end{array}$$

After the above transformation, we have:(33)$$\begin{array}{c} {\hat{C}}_{l,u}^{*}=\Xi_{l,u}\left[\xi_{l,1}\left(\Omega_{u}\right)-\xi_{l,2}\left(\Omega_{u}\right)\right]+\Psi_{l,u}, \end{array}$$
(34)$$\begin{array}{c} {\hat{C}}_{m,u}^{*}=\Xi_{m,u}\left[\xi_{m,1}\left(\Omega_{u}\right)-\xi_{m,2}\left(\Omega_{u}\right)\right]+\Psi_{m,u}, \end{array}$$

Next, we adopt DC programming to make the problem standard semi-definite programming and then use the MOSEK toolbox of MATLAB for obtaining an efficient solution.

## 4. Numerical Results and Discussion

The numerical results of the presented optimization scheme are provided here. Parameters for the simulation are as follows, unless otherwise specified: All drones use the same frequency for optimal spectrum utilization, the value of the imperfect SIC is set to 0.1, the variance of AWGN is set to 0.01, the average channels are obtained from $10^{3}$ realizations, the transmit power of each drone is 30 dBm, the passive elements of each IRS are set to 50, the number of ground IoT devices associated with each drone is two, and the path loss exponent is three. In addition, the system parameters are also provided in Table 2.

To begin with, it is of the utmost significance to have demonstrated the complexity of the proposed IRS-assisted PD-NOMA drone communication scheme. Figure 2 illustrates the behavior of convergence for the total achievable capacity as a function of the number of iterations in this regard. It is clear from looking at the figure that the system converges within a fair number of iterations, and that increasing the number of drones that are interfering has little effect on the amount of time it takes for the system to converge.

Following that, it is critical to demonstrate the effect of drone transmit power on the system’s total achievable capacity. Figure 3 depicts the increasing values of each drone power versus the system’s total achievable capacity for different number of drones and minimum capacity requirements. The total achievable capacity grows as the drone’s transmit power is increased. Furthermore, as more available transmit power at the drone becomes available for optimization, the gap in the total achievable capacity offered by all scenarios grows. As a result, the scenario with fewer restrictions on capacity requirements outperforms other scenarios with higher capacity requirements. Aside from that, we can see that the total achievable capacity increases in a similar manner as the number of drones increases from five to ten. This validates the proposed scheme for a large PD-NOMA IRS-assisted drone communications network.

To examine the impact of SIC errors on capacity when IoT devices decode their signals, Figure 4 plots the system’s total achievable capacity against increasing SIC error values where the number of drones is five and ten and the minimum capacity requirement of each IoT device is set at 0.5 b/s/Hz and 1 b/s/Hz. As the value of SIC errors increases, the total achievable capacity of all considered scenarios is also decreased. Because of the poor signal decoding capability, high values of SIC errors increase interference among IoT devices on the same frequency. Another point to mention is the decreasing capacity gap of all scenarios as SIC errors increase. This is because, as SIC errors increase, more power is required by all IoT devices to meet the minimum capacity requirements. As a result, the power distribution becomes less flexible. Obtaining perfect SIC in practical systems is a difficult task. However, most works in the literature have considered perfect SIC for ease of solution. As we can see, using perfect SIC results in an overly optimistic performance evaluation.

Figure 5 depicts the total achievable capacity of the system versus the passive elements of each IRS for a number of different drones and minimum capacity requirements. In both scenarios, we can see that with the increasing number of IRS passive elements, the total achievable capacity also increases. It can also be seen that when the capacity requirements are high, the total achievable capacity is lower than when the capacity requirements are low. The key concept here is that in high capacity requirements scenarios, some IoT devices struggle to meet their capacity requirements. In such scenarios, IoT devices with high capacity reduce their power to minimize interference received by weak IoT devices and assist them in meeting their capacity requirements. Although IoT devices meet their minimum capacity requirements, the overall system capacity decreases. Furthermore, the figure shows that the system with ten drones achieves nearly double the total achievable capacity as the system with five drones. This shows how important the proposed scheme is for PD-NOMA IRS-enhanced drone communications networks on a large scale.

Figure 6 depicts the effect of IRS elements on system performance. Here, we plot the system’s total achievable capacity versus an increasing number of IRS elements. We compared the performance of the considered system model to a scenario in which the IRS only smartly reflects the signal of far IoT devices, while the strong IoT devices receive their signals directly from drones. As expected, increasing the number of IRS elements increases the system’s total achievable capacity for both scenarios. Furthermore, we can see that when each IRS reflects the data of both IoT devices, we get much better performance. All of the results in this section showed that systems with a greater number of drones have a greater achievable capacity. This demonstrates the significance of the proposed scheme for large-scale PD-NOMA IRS-assisted drone communications networks.

## 5. Conclusions

The incorporation of PD-NOMA and IRS in multi-drone communications has the potential to extend wireless communication and connect large numbers of devices in future wireless networks. This paper proposes a new optimization scheme for PD-NOMA IRS-assisted drone communication networks in order to maximize the system’s total achievable capacity. Specifically, under the assumption of SIC errors, our framework simultaneously optimizes the transmission power of each drone and the reflection coefficient of each IRS. KKT conditions were used to calculate the total transmission power of drones and the power allocation coefficients of IoT devices. Using successive convex approximation and DC programming, an efficient IRS reflection coefficient was then designed. The benefits of the proposed optimization scheme have been demonstrated by simulation results. 

## Figures and Tables

**Figure 1 sensors-22-08603-f001:**
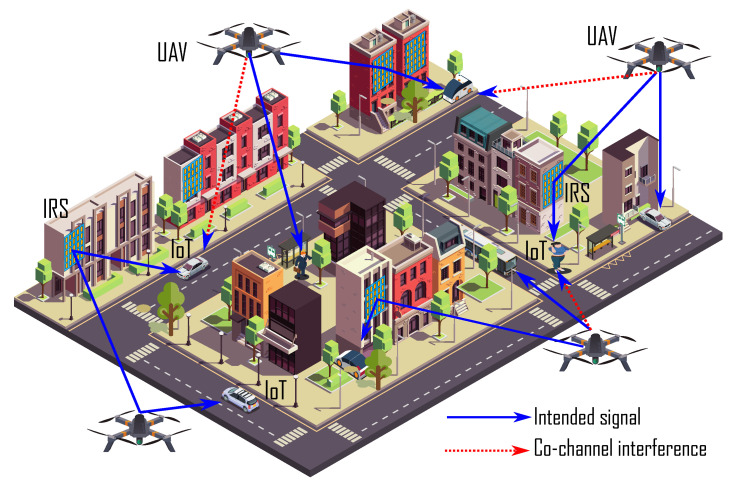
NOMA IRS-assisted drone communications.

**Figure 2 sensors-22-08603-f002:**
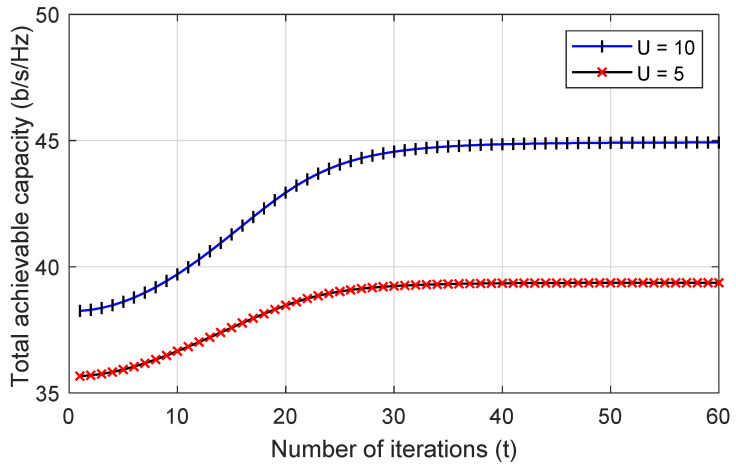
Convergence of the proposed PD-NOMA IRS-enhanced drone communications scheme.

**Figure 3 sensors-22-08603-f003:**
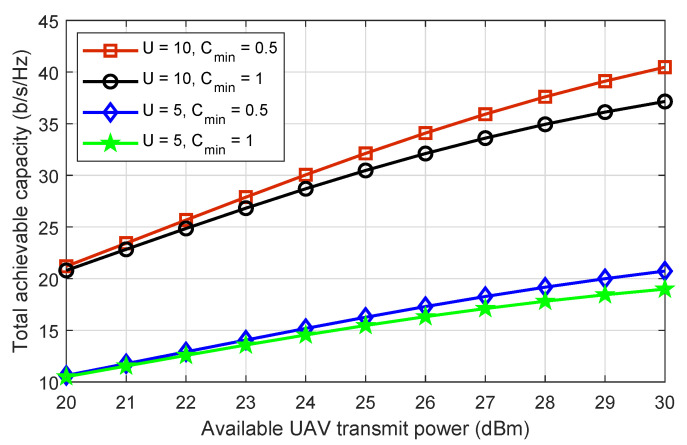
The total achievable capacity of the system versus the available transmit power of each drone, where the number of drones are five and ten while the minimum capacity requirements are set as 0.5 and 1 b/s/Hz.

**Figure 4 sensors-22-08603-f004:**
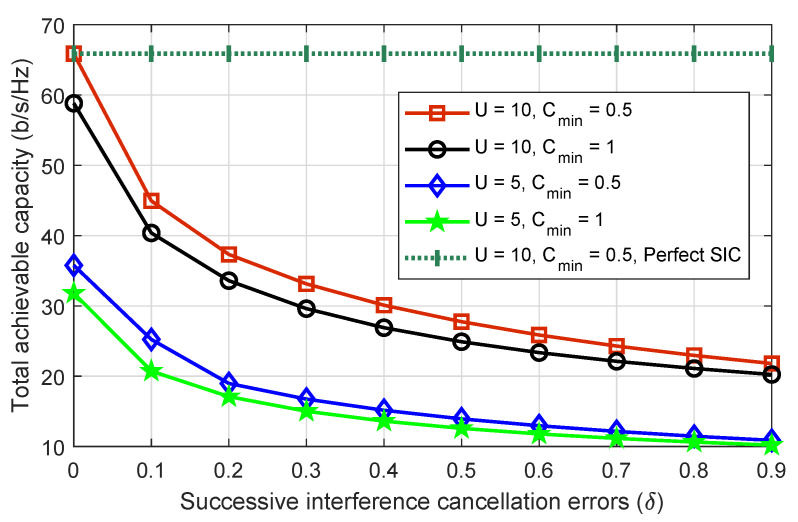
The total achievable capacity of the system versus increasing SIC errors, where the number of drones are five and ten while the minimum capacity requirements are set as 0.5 and 1 b/s/Hz.

**Figure 5 sensors-22-08603-f005:**
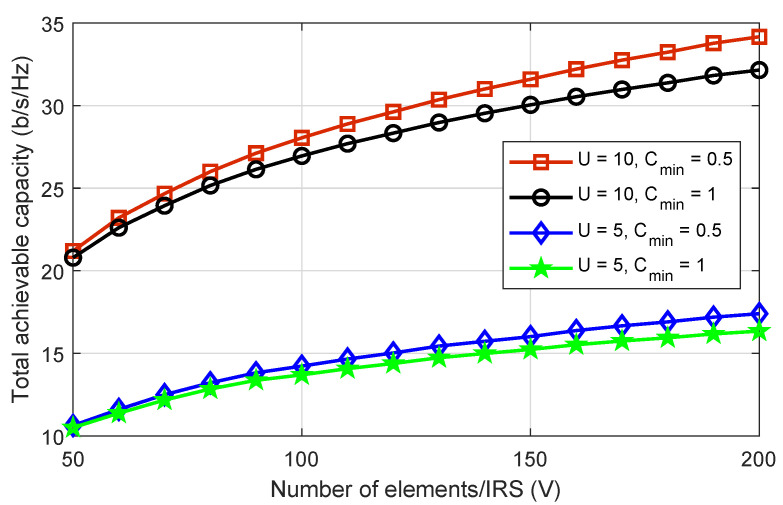
The total achievable capacity of the system versus the number of passive elements of IRS, where the number of drones are five and ten while the minimum capacity requirements are set as 0.5 and 1 b/s/Hz.

**Figure 6 sensors-22-08603-f006:**
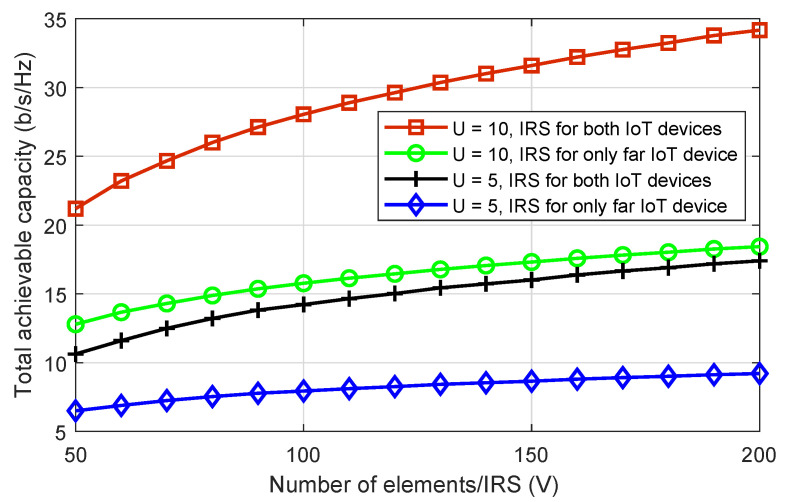
The total achievable capacity of the system versus the number of passive elements of IRS, where the number of drones are five and ten.

**Table 1 sensors-22-08603-t001:** List of notations used in this work.

Notation	Definition
*U*	Set of drones
*u*	Drone index
Θ	Phase shift matrix of IRS
ψ	Amplitude of passive reflection of IRS element
θ	Phase shift of IRS element
*x*	Transmitted superimposed signal of drone
*v*	Passive element of IRS
l,m	Index IoT devices
*Q*	Power budget of drone
ϱ	PD-NOMA power allocation coefficient
*h*	Channel between drone and IRS
ϑ	Reference channel gain over 1 meter
ζ	2D coordinate of drone
υ	Location of IRS and IoT devices on horizontal plane
*H*	Altitude of drone
*g*	Channel between IRS and IoT devices
*G*	Rayleigh fading coefficient
*D*	Distance between IRS and IoT device
*y*	Received signal at IoT device
ω	Additive white Gaussian noise
*C*	Capacity of IoT device
Γ	Co-channel interference
$C_{min}$	Minimum capacity of IoT devices
$P_{max}$	Maximum power budget of IoT device
L	Lagrangian function
λ	Vector of Lagrangian multipliers
μ	Lagrangian multiplier

**Table 2 sensors-22-08603-t002:** System parameter details.

Parameter	Definition
Number of drones	10
Number of IoT devices	10
Number of IRSs	10
Imperfect SIC parameter	0.1
Monte Carlo simulation	1000
IRS passive elements	50
Power budget of each drone	30
Path loss exponent	3
Additive white Gaussian noise	0.01
Altitude of drone	80

## Data Availability

Not applicable.

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
