# Peer review of "Joint Power Control and Phase Shift Design for Future PD-NOMA IRS-Assisted Drone Communications under Imperfect SIC Decoding"

_sensors, 2022, doi:10.3390/s22228603_

Round 1

Reviewer 1 Report

The presented paper is devoted to the well-known issues related to the optimization of communication techniques using NOMA and intelligent reflecting surfaces (IRSs). The authors of the paper prepared a very synthetic introduction to the problematics related to the UAV communications. The state-of-the-art has been conducted quite widely, but it is worth putting more emphasis on the analysis of the references strictly devoted to common optimization in the field of access techniques based on power and space management. The reviewer believes that the presented paper can be successfully published after introducing a few simple corrections and extensions listed below:

11. The format of the document is inconsistent with that required by the publishing house and the journal.

22.  It is worth adding a short description of NOMA and IRS with the classification of solutions.

33.  Examples of real conditions enabling the use of NOMA and IRS techniques with the parameters determined during the simulation, i.e. operating frequency, drone requirements for access to bit-rate, the impact of the complexity of the propagation environment other than the one proposed with the exponent 3, the impact of other modulation techniques on the functioning of NOMA in combination with IRS, the use of MIMO in drones in combination with time-space coding depending on the structure and modularity of the IRS, should be presented.

44. The obtained results should be discussed in relation to other solutions proposed in the literature (extended state-of-the-art).

55.  Please introduce simple text corrections in the paper:   

a.       Line 1 – correct abbreviation - “Intelligent reflecting surfaces (IRSs)”.

b.       Line 19 – correct keyword – “non-orthogonal multiple access“.

c.       Line 28 – correct abbreviation – “power domain non-orthogonal multiple access (PD-NOMA)”, but second abbreviation is not needed.

d.       DC and UAV abbreviations are missed in the body of presented paper. We can find them in the references only.

e.       Below formula (3) the reference is missed: Next, following [? ].

f.        Line 159 and below line 162 – reference (18) is incorrect. Shouldn't there be (9)?

g.       Line 201 – figure number is missed.

Author Response

Reviewer 1

  1. The presented paper is devoted to the well-known issues related to the optimization of communication techniques using NOMA and intelligent reflecting surfaces (IRSs). The authors of the paper prepared a very synthetic introduction to the problematics related to the UAV communications. The state-of-the-art has been conducted quite widely, but it is worth putting more emphasis on the analysis of the references strictly devoted to common optimization in the field of access techniques based on power and space management. The reviewer believes that the presented paper can be successfully published after introducing a few simple corrections and extensions listed below:

Response: Thank you very much for the positive comments and suggestions. Below are our responses to your comments.

  1. The format of the document is inconsistent with that required by the publishing house and the journal.

Response: Thank you so much for carefully reviewing our manuscript. In this regard we would like to clarify that the template we are using for this paper is Latex version and has been downloaded from the journal site. Note that we have used the same template before for publication in MDPI and they do not have any problem with the production team.

  1. It is worth adding a short description of NOMA and IRS with the classification of solutions.

Response: Thank you so much for your comments. We have now added another paragraph on NOMA and IRS with the classification of solution. Please refer to the revised paper.

  1. Examples of real conditions enabling the use of NOMA and IRS techniques with the parameters determined during the simulation, i.e. operating frequency, drone requirements for access to bit-rate, the impact of the complexity of the propagation environment other than the one proposed with the exponent 3, the impact of other modulation techniques on the functioning of NOMA in combination with IRS, the use of MIMO in drones in combination with time-space coding depending on the structure and modularity of the IRS, should be presented.

Response: Thank you so much for you comments. We appreciate this reviewer for technical reviewing our manuscript. We have now explained in more detail the parameters used in this paper. This work considered link level simulation to check the performance of NOMA and IRS in UAV networks. More practical system can be considered in future such that this work should be treated as a benchmark scheme.

(Asad bhai: I have added notations table, please add simulation parameters table.)

  1. The obtained results should be discussed in relation to other solutions proposed in the literature (extended state-of-the-art).

Response: Thank you so much for your comments. Due to novelty of our system model, it is difficult to directly compare this work with the existing literature. Therefor, we resort to compare the proposed technique with suboptimal techniques, i.e., proposed technique without IRS, proposed technique with perfect SIC decoding, and proposed technique where only far user signal is assisted by IRS.

  1. Please introduce simple text corrections in the paper:   
  2. Line 1 – correct abbreviation - “Intelligent reflecting surfaces (IRSs)”.
  3. Line 19 – correct keyword – “non-orthogonal multiple access“.
  4. Line 28 – correct abbreviation – “power domain non-orthogonal multiple access (PD-NOMA)”, but second abbreviation is not needed.
  5. DC and UAV abbreviations are missed in the body of presented paper. We can find them in the references only.
  6. Below formula (3) the reference is missed: Next, following [? ].
  7. Line 159 and below line 162 – reference (18) is incorrect. Shouldn't there be (9)?
  8. Line 201 – figure number is missed.

Response: Thank you very much for carefully reviewing our paper. An extensive proofreading has been done for typos and grammar errors and correct those where needed. Please refer to the revised manuscript.

Reviewer 2 Report

This paper aims to enhance the total spectral efficiency by optimizing the power budget of each drone, PD-NOMA power allocation of IoT devices, and the phase shift matrix of IRS. The paper is well presented but there are some observations as follows.

1.     Although the authors have mentioned the objective of the paper, but it will be better for the readers if the authors point-wise present the significant contribution of the paper.

2.     Regarding the system model, the mathematical model suggests that it’s a Single IRS and multiple UAVs but the pictorial representation is something different. Please follow the consistency. Otherwise mention that selected IoT devices are supported by only one IRS.

3.     Eqn. 7.. if it is the author's own contribution, please provide in detail else provide proper citation.

4.     Please include the explanation corresponding to Eq. 9.

5.     Refer to Figure 1, in the text.

6.     Page 11…Line 201… Please correct figure no.

7.     Page 5…..Next, the following [? ],

8.     Thorough proofreading is required.

Author Response

Reviewer 2

  1. This paper aims to enhance the totalspectral efficiency by optimizing the power budget of each drone, PD-NOMA power allocation of IoT devices, and the phase shift matrix of IRS. The paper is well presented but there are some observations as follows.

Response: We are very thankful to this reviewer for giving positive decision. Below are our responses to your comments.

  1. Although the authors have mentioned the objective of the paper, but it will be better for the readers if the authors point-wise present the significant contribution of the paper.

Response: Thank you so much for your valuable comments. We have now rewritten the main motivation and contributions of this paper. Specifically, we have presented our contributions pointwise in the revised manuscript. Please refer to the revised paper.

  1. Regarding the system model, the mathematical model suggests that it’s a Single IRS and multiple UAVs, but the pictorial representation is something different. Please follow the consistency. Otherwise mention that selected IoT devices are supported by only one IRS.

Response: Thank you so much for your comments. In this regards we would like to clarify that we consider multiple UAVs and multiple IRSs. In each coverage area of UAV, one IRS is mounted on strategic position to assist the signal delivery from UAV to ground users. Yet to address these comments, we have now revised the system model in more details. Please refer to the revised manuscript.

  1. 7, if it is the author's own contribution, please provide in detail else provide proper citation.

Response: Thank you so much for you comments. We have now explained Eq. 7 and 8 in more details for better understanding of the reader. Please refer to the revised paper.

  1. Please include the explanation corresponding to Eq. 9.

Response: Thank you so much for your comments. Eq. 9 is the objective function of sum capacity maximization. We have now added the explanation of Eq. 9 in the revised paper.

  1. Refer to Figure 1, in the text.

Response: Thank you so much for carefully reviewing our paper and identifying this mistake. We have now referred Figure 1 in the text. Please refer to the first paragraph of the system model and problem formulation section.

  1. Page 11…Line 201… Please correct figure no.

Response: Thank you very much for your comments. We have corrected figure number as “Figure 5”. Please refer to the revised manuscript.

  1. Page 5, Next, the following [? ],

Response: Thank you for identifying this typo, we have now corrected it in the revised version of our paper.

  1. Thorough proofreading is required.

Response: Thank you so much for your comments. An extensive proofreading has been done for typos and grammar errors and corrected those where needed.

Reviewer 3 Report

Title: Joint Power Control and Phase Shift Design for Future PD-NOMA IRS-Assisted Drone Communications under Imperfect SIC Decoding

Comments: The work is interesting and timely. However, the authors may address the following comments to improve the quality further.

1. The introduction section is not able to define the objective and state of art. The authors may create a subsection or highlight the objectives (pointwise) and state of art.

2. Some of the statements need in-depth discussions.

a. Following the PD-NOMA principle, the IoT device l associated with UAV u can apply SIC to decode the signal of IoT device m before decoding its own signal. However, we assume that IoT device l cannot always decode it successfully.

b. Constraint W3 control the transmit power of each drone, where Pmax shows the 156 maximum power budget. 

c. To handle the complexity of this problem and obtain efficient solution, we exploit successive convex approximation method.

d. We can see that as the value of SIC errors increases, the total achievable capacity 195 decreases for all scenarios considered.

3. Recent works on drone/UAV-based communication may be discussed in Section 1.1. ex: "Multi-antenna relay based cyber-physical systems in smart-healthcare NTNs: an explainable AI approach".

4. The simulation scenarios need more discussions along with their limitations of them.

5. The authors can add a notation table and simulation scenario table for better understanding.

Author Response

Reviewer 3

  1. The work is interesting and timely. However, the authors may address the following comments to improve the quality further.

Response: Thank you so much for reviewing our paper and giving us a productive decision and comments. Below are our responses.

  1. The introduction section is not able to define the objective and state of art. The authors may create a subsection or highlight the objectives (pointwise) and state of art.

Response: Thank you very much for your comments. We have now added subsections in Introduction section for literature review and motivation and contribution. Please refer to the revised paper.

  1. Some of the statements need in-depth discussions.

Response: Thank you so much for your comments. In this regards we have revised all the below statements and write it in more clear way.

  1. Following the PD-NOMA principle, the IoT device l associated with UAV u can apply SIC to decode the signal of IoT device m before decoding its own signal. However, we assume that IoT device l cannot always decode it successfully.
  2. Constraint W3 control the transmit power of each drone, where Pmax shows the 156 maximum power budget. 
  3. To handle the complexity of this problem and obtain efficient solution, we exploit successive convex approximation method.
  4. We can see that as the value of SIC errors increases, the total achievable capacity 195 decreases for all scenarios considered.

Response: Thank you so much for your comments. In this regards we would like to clarify that based on downlink NOMA principle, near user can apply SIC to decode the signal of far user before decoding its own signal. However, it is hard to decode it perfectly, thus, we consider that near user might not be capable to decode the signal of far user perfectly and there exist an error in SIC decoding process. Yet to address these comments, we have also revised the text in the paper. Please refer to the revised manuscript.

W3 shows controls the transmit power of each drone, where Pmax shows the maximum power budget of drones.

To reduce the complexity of the above problem and obtain efficient solution, we exploit successive convex approximation method.

As the value of SIC errors increases, the total achievable capacity of all considered scenarios is also decreased.

  1. Recent works on drone/UAV-based communication may be discussed in Section 1.1. ex: "Multi-antenna relay based cyber-physical systems in smart-healthcare NTNs: an explainable AI approach".

Response: Thank you so much for your comments. The above reference has been studied and reported in the revised manuscript. Please refer to the paper.

  1. The simulation scenarios need more discussions along with their limitations of them.

Response: Thank you so much for your comments. We have now discussed all the considered scenarios in more detail. Please refer to the simulation and discussion section.

  1. The authors can add a notation table and simulation scenario table for better understanding.

Response: Thank you so much for your comments. For the readers of this work, we have now added notations and simulation parameters tables. Please refer to revised manuscript.

Round 2

Reviewer 3 Report

The author's have incorporated all the queries. No further questions from my side.